# Unleashing Foundation Vision Models: Adaptive Transfer for Diverse Data-Limited Scientific Domains

**Qiankun Li**[*1,3], **Feng He**[1], **Huabao Chen**[1], **Xin Ning**[2], **Kun Wang**[*3], **Zengfu Wang**[*1]

[1]University of Science and Technology of China
[2]AnnLab, Institute of Semiconductors, Chinese Academy of Sciences
[3]Nanyang Technological University
*Corresponding Author
{qklee,wk520529}@mail.ustc.edu.cn, zfwang@ustc.edu.cn

## Abstract

In the big data era, the computer vision field benefits from large-scale datasets such as LAION-2B, LAION-400M, and ImageNet-21K, Kinetics, on which popular models like the ViT and ConvNeXt series have been pre-trained, acquiring substantial knowledge. However, numerous downstream tasks in specialized and data-limited scientific domains continue to pose significant challenges. In this paper, we propose a novel Cluster Attention Adapter (CLAdapter), which refines and adapts the rich representations learned from large-scale data to various data-limited downstream tasks. Specifically, CLAdapter introduces attention mechanisms and cluster centers to personalize the enhancement of transformed features through distribution correlation and transformation matrices. This enables models fine-tuned with CLAdapter to learn distinct representations tailored to different feature sets, facilitating the models' adaptation from rich pre-trained features to various downstream scenarios effectively. In addition, CLAdapter's unified interface design allows for seamless integration with multiple model architectures, including CNNs and Transformers, in both 2D and 3D contexts. Through extensive experiments on 10 datasets spanning domains such as generic, multimedia, biological, medical, industrial, agricultural, environmental, geographical, materials science, out-of-distribution (OOD), and 3D analysis, CLAdapter achieves state-of-the-art performance across diverse data-limited scientific domains, demonstrating its effectiveness in unleashing the potential of foundation vision models via adaptive transfer. Code is available at https://github.com/qklee-lz/CLAdapter.

## 1 Introduction

With the rapid advancement in artificial intelligence, deep learning-based computer vision algorithms have emerged as a dominant force [42, 85, 47]. These algorithms are inherently data-driven, capitalizing on substantial datasets to refine their task-specific performance. The digital age's ever-growing data trove has ushered in large-scale datasets, such as ImageNet-21K [58], LAION-400M [62], and LAION-2B [61], which aim to bolster algorithmic generalization and accuracy through data diversity and volume [25, 44, 10]. Despite these advancements, domain-specific challenges and data scarcity remain significant hurdles in scientific visual downstream tasks, where specialized data is often limited, heterogeneous, or expensive to acquire [11, 72, 79]. Therefore, developing methods that effectively harness the potential of large-scale pre-trained models to enable robust adaptation in data-limited scientific domains constitutes a critical and promising research direction.

Transfer learning through methods like linear probing and full fine-tuning is an essential approach for enhancing performance on downstream tasks [77]. This works well when transferring under

39th Conference on Neural Information Processing Systems (NeurIPS 2025).

normal-sized dataset pre-trained models to in-distribution (ID) downstream tasks. However, transferring adapted knowledge from rich but complex large-scale upstream pretraining poses significant challenges in the current era of large datasets [61, 10]. Furthermore, the scientific domains downstream tasks often involve out-of-distribution (OOD) scenarios [4], domain specificity [27], and data limitations [68], intensifying the fine-tuning challenge. L2-SP fine-tuning [26] introduced L2 regularization to preserve the insights of pre-trained weights during task adaptation, although they may lack robustness in OOD contexts. Visual Prompt Tuning (VPT) [34] augmented the input space with task-specific learnable prompts. However, VPT is primarily designed for Vision Transformer (ViT) [70] and lacks the ability to provide stable cross-domain feature transferability. OLOR [30] designed the fine-tuning optimizer from the perspective of pre-trained weights to improve stability, but it lacks task-specific self-adaptability, particularly under the diverse conditions of downstream scientific tasks.

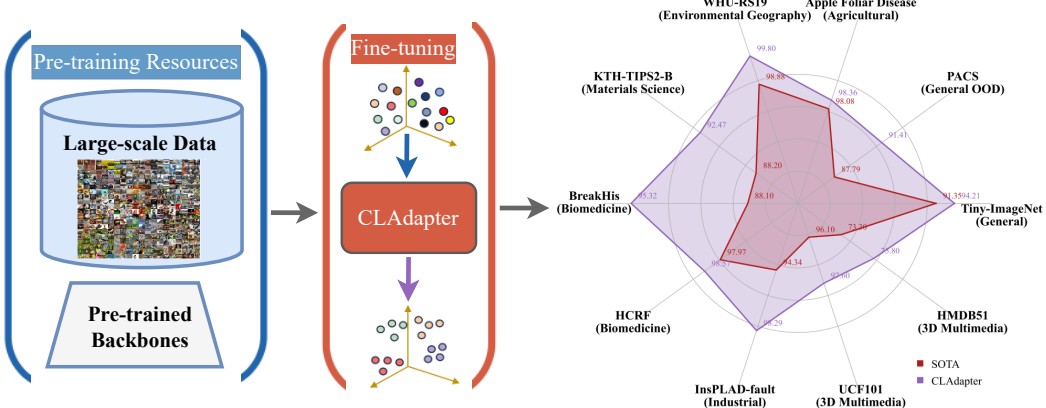

Figure 1: Overview of the proposed CLAdapter. CLAdapter refines and adapts the rich representations learned from large-scale data to diverse data-limited scientific downstream tasks, achieving state-of-the-art performance across diverse fields on 10 datasets.

In this paper, we propose a novel Cluster Attention Adapter (CLAdapter), which refines and adapts the rich representations learned from large-scale data to diverse data-limited scientific downstream tasks (as illustrated in Figure 1). Specifically, CLAdapter introduces attention mechanisms and cluster centers to enable customized feature enhancement through distribution-aware correlation and transformation matrices. This facilitates the generation of task-adaptive representations, supporting a smooth transition from abundant pre-trained features to diverse downstream scenarios. Benefiting from our unified interface design, CLAdapter can seamlessly integrate with mainstream architectures, including CNNs, Transformers, and their 3D versions. In addition, a Staged Fine-Tuning (SFT) strategy is presented to collaborate with CLAdapter to further enhance the fine-tuning performance. Through extensive experiments conducted on 10 datasets spanning domains such as generic, multimedia, biological, medical, Industrial, agricultural, environmental, geographical, materials science, out-of-distribution (OOD), and 3D analysis, CLAdapter demonstrates its universal applicability and state-of-the-art performance. These results underscore the importance of effective knowledge transfer in the big data era and advance the reliable and efficient deployment of computer vision foundation models across scientific and industrial domains.

The contributions of this paper are summarized as follows:

① **Adaptive Representation Transfer.** We propose a novel CLAdapter that leverages large-scale pre-trained knowledge to enhance performance on a variety of data-limited downstream tasks.

② **Flexible Adaptation Framework.** We design a unified interface and a staged fine-tuning (SFT) strategy, enabling CLAdapter to integrate seamlessly with mainstream pre-trained models and establish an efficient fine-tuning paradigm.

③ **AI4Science Broad Evaluation.** We conduct comprehensive experiments on 10 datasets across diverse domains, including multiple scientific fields where data is limited and heterogeneous. CLAdapter consistently achieves state-of-the-art performance, demonstrating its potential as a generalizable solution for AI-driven scientific applications.

## 2 Related Work

### 2.1 Pre-Training Resources

With the rapid development of computer vision technology, a large number of large-scale datasets [58, 62] and pre-trained models [56, 57, 5, 25, 19] have been proposed, providing a rich feature library for downstream tasks. Pre-training on large-scale datasets can encode rich semantic information, which is useful in solving limited data tasks, domain generalization, and zero-shot learning. ImageNet-21k[58], mainly used for visual image classification, contains about 21k categories and 14 million images, providing a rich and diverse training environment for large models. The LAION-400M dataset [62] is a large-scale image and text pairing dataset, containing about 300 million image-text pairs, including natural landscapes, people, everyday items, etc., suitable for training cross-modal models [56, 57]. To cope with these growing resources, a new generation of pre-trained models (such as CLIP [56], BEiT [5], MAE [25], and EVA [19]) has emerged, mainly utilizing the architectural principles of ViT [70] and ConvNeXt [50]. However, how to efficiently transfer a large number of complex datasets remains an unresolved issue. Therefore, this paper proposes CLAdapter, introducing attention mechanisms and clustering centers to leverage rich pre-training resources, thus effectively improving performance on various downstream tasks.

### 2.2 Various Downstream Tasks and Fine-Tuning Methods

The realm of practical downstream visual tasks is vast. However, most are cross-domain with limited data, such as medical image processing [11], industrial fault diagnosis [60], pest and disease recognition [72], and natural geographic image classification [83], Materials science research [45], 3D multimedia analysis [37]. Fine-tuning pre-trained models has become a critical method for improving performance on downstream tasks. The popular fine-tuning methods are Linear Probing (LP) [26], adjusting only the model's head, and Full Fine-tuning (FT), tuning all layers. Recently, L2-SP fine-tuning introduced an L2 regularization to keep changes to pre-trained weights minimal, thus preserving initial insights while adapting to new tasks. Visual Prompt Tuning (VPT) [34] inserted trainable prompts at the input, akin to NLP's prompt learning, to prevent altering the original model weights. VPT proved effective for tasks with numerous parameters and scant data, utilizing pre-trained knowledge and averting overfitting from significant weight modifications. However, VPT is mainly designed for Vision Transformer (ViT) [17] and has cross-domain instability issues. Different from existing fine-tuning methods, CLAdapter introduces attention mechanisms and cluster centers for customized feature representation refinement. By providing a uniform interface for various upstream tasks, CLAdapter fine-tunes any category of pre-trained models, transferring their knowledge to a wide array of downstream tasks.

## 3 Methods

We propose CLAdapter to refine and adapt the rich representations learned from large-scale data for transfer applications in various downstream tasks. It injects a small number of learnable parameters into the original model, offering an efficient fine-tuning strategy by freezing the backbone, alongside a staged fine-tuning approach for more gradual adaptation. The framework is depicted in Figure 2.

### 3.1 Unified Model Interface

Given an input image $X_I \in \mathbb{R}^{C \times H \times W}$, a standard Vision Transformer (ViT) divides $X_I$ into $N$ patches. Each patch is then embedded into a $D$-dimensional latent space, resulting in a set of tokens $\mathcal{T} \in \mathbb{R}^{N \times D}$. Then, $\mathcal{T}$ and a extra learnable classification token $x_{class}$ ([$class$]) with position embeddings are fed into the transformer layers $\{E^l\}_{l=0}^{L-1}$. Thus, the feature representation extracted by ViT is $\mathcal{T}^L \in \mathbb{R}^{(N+1) \times D}$. Discarding $x_{class}$ due to its linear combination nature, as it might interfere with feature transfer from the pre-train model to downstream tasks.

For CNN-based models, the feature map $X_F \in \mathbb{R}^{C' \times H' \times W'}$ is extracted. Here, $C'$, $H'$, and $W'$ represent the number of channels, height, and width of the feature map, respectively. To facilitate the integration of CNN-based models with CLAdapter, we flatten the spatial dimensions of the feature map $X_F$ to align with the feature dimensionality used by ViT.

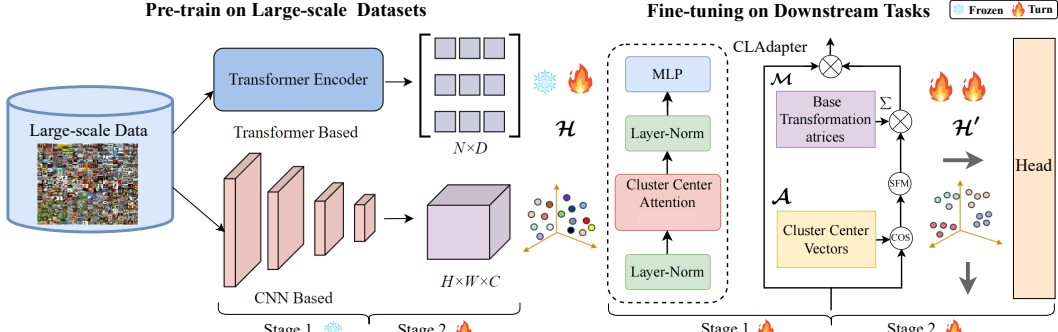

Figure 2: Overview of the CLAdapter. It utilizes large-scale pre-training to enhance various data-limited downstream tasks. The unified interface design and SFT fine-tuning strategy allow CLAdapter to integrate with mainstream pre-trained models and form an efficient fine-tuning paradigm.

In the case of 3D video clips or image sequences $\boldsymbol{X}_V \in \mathbb{R}^{T \times C \times H \times W}$, the extracted features also include an additional temporal dimension $T$. Although time and spatial dimensions together form tubes rather than patches, they still belong to the internal structure of data. Therefore, we similarly flatten these dimensions to achieve a unified representation.

In summary, whether for CNN-based models, Transformer-based models, or their 3D variants, we uniformly apply an interface function for the features extracted by the models, denoted as $\mathcal{F}(\boldsymbol{X}_I/\boldsymbol{X}_V) \to \mathcal{H}$, where $\mathcal{H} \in \mathbb{R}^{N \times D}$ represents the extracted features after dimension unification.

## 3.2 Cluster Attention Adapter (CLAdapter)

For downstream tasks, it is crucial to focus on specific information pertinent to their domain. However, while large-scale upstream data encompasses a wealth of information, it also introduces complexity and redundancy. This challenge is further exacerbated when dealing with Out-Of-Distribution (OOD) tasks, making it more difficult to distill the necessary information from the rich pre-trained feature representations $\mathcal{H}$. In real-world applications, with the diverse nature of downstream tasks presenting both in-distribution (ID) and OOD scenarios, it is crucial to design a mechanism capable of adaptively refining and transforming the pre-trained features $\mathcal{H}$ into suitable features $\mathcal{H}'$ based on the characteristics of the downstream tasks. This process can be defined as learning a mapping function $\mathcal{F}_\theta(\mathcal{H}) \to \mathcal{H}'$, where $\mathcal{F}_\theta$ denotes a model and $\boldsymbol{\theta}$ represents its learnable parameters.

CLAdapter aims to refine and adapt the feature representations $\mathcal{H}$ learned from large-scale datasets for use in various data-limited downstream tasks. Notably, embeddings of image categories in feature space are often close to each other, suggesting the presence of feature cluster centers that represent specific latent information. To exploit this, we introduce multiple learnable vectors to denote these feature cluster centers:

$$\mathcal{A} = \{\mathcal{A}_1, \mathcal{A}_2, \cdots, \mathcal{A}_K\}, \tag{1}$$

where $\mathcal{A} \in \mathbb{R}^{D \times K}$, and $K$ is the number of cluster centers. The attention scores are derived by calculating the cosine similarity between the pre-trained embeddings $\mathcal{H}$ and the cluster centers $\mathcal{A}$. To enhance computational efficiency and convert attention scores to a relative probability distribution $\boldsymbol{\beta}$, we compute the mean of $\mathcal{H}$ to obtain $\mathcal{H}^q$ and apply the softmax function, respectively. The above operation is denoted mathematically as follows:

$$\mathcal{H} = LayerNorm(\mathcal{H}), \quad \mathcal{H}^q = \frac{1}{N} \sum_{i=1}^{N} \mathcal{H}_i, \tag{2}$$

$$\hat{\mathcal{H}}^q = \frac{\mathcal{H}^q}{\|\mathcal{H}^q\|}, \quad \hat{\mathcal{A}} = \left[ \frac{\mathcal{A}_1}{\|\mathcal{A}_1\|}, \frac{\mathcal{A}_2}{\|\mathcal{A}_2\|}, \cdots, \frac{\mathcal{A}_K}{\|\mathcal{A}_K\|} \right], \tag{3}$$

$$\boldsymbol{\beta} = \text{softmax}\left( \hat{\mathcal{H}}^q \hat{\mathcal{A}} \right), \tag{4}$$

where Layer-Norm reduces the difference in input distribution between different layers, and both $\hat{\mathcal{H}}^q$ and $\hat{\mathcal{A}}$ are L2 normalized along the feature dimension. Upon obtaining the attention scores $\boldsymbol{\beta} \in \mathbb{R}^K$,

we further introduce learnable transformation matrices $\mathcal{M}$ corresponding to the cluster centers $\mathcal{A}$:

$$\mathcal{M} = \{\mathcal{M}_1, \mathcal{M}_2, \cdots, \mathcal{M}_K\}, \tag{5}$$

where each transformation matrix $\mathcal{M}_i \in \mathbb{R}^{D \times D}$. The weighted transformation matrix $\mathcal{M}^*$ for each pre-trained feature embedding is derived by weighting these matrices with the attention scores:

$$\mathcal{M}^* = \sum_{i=1}^{K} \beta_i \mathcal{M}_i. \tag{6}$$

Consequently, each embedding is subjected to a custom transformation matrix based on its cosine similarity with each feature cluster center, facilitating the organized transition from the original upstream feature distribution to a new distribution tailored for downstream tasks. This involves a customized transformation of $\mathcal{H}$ using $\mathcal{M}^*$, followed by enhancement with a Layer-Norm and MLP layer to improve generalization and introduce non-linearity. The above operation is denoted mathematically as follows:

$$\mathcal{H}^* = LayerNorm(\mathcal{H}\mathcal{M}^*), \tag{7}$$

$$\mathcal{H}' = GELU(\mathcal{H}^*\boldsymbol{W}_1 + \boldsymbol{b}_1)\boldsymbol{W}_2 + \boldsymbol{b}_2, \tag{8}$$

where $\mathcal{H}^*$ is the result of the customized transformation. In the MLP, $GELU$ represents the Gaussian Error Linear Unit activation function, and $\boldsymbol{W}_1$ and $\boldsymbol{W}_2$ are the weight matrices for the first and second linear transformations, respectively, with a default ratio of 4. $\boldsymbol{b}_1$ and $\boldsymbol{b}_2$ are the corresponding bias vectors. The final output $\mathcal{H}'$ represents the features adaptively refined and transformed from the rich pre-trained feature embeddings $\mathcal{H}$ to suit downstream tasks. Additionally, these transformed features can be reshaped back to the original feature shape of the upstream model through inverse function of the unified interface defined in Section 3.1, facilitating the integration with their respective heads.

## 3.3 Fine-tuning Strategy

To adapt a pre-trained model for a downstream task, practitioners commonly employ either full fine-tuning (FT), where all model parameters are updated, or linear probing (LP), which only updates the parameters of the final linear classification layer (head). By incorporating our proposed CLAdapter, the LP approach updates both the adapter and the classification layer, whereas the FT approach updates the entire model. These two popular fine-tuning strategies can be formalized as:

$$\mathcal{LP}_{CL}(x) = \boldsymbol{W}_{hd} \cdot h_{CL}(\mathcal{H}) + \boldsymbol{b}_{hd}, \tag{9}$$

$$\mathcal{FT}_{CL}(x) = \boldsymbol{W}_{hd} \cdot h_{CL}(\phi_{pre}(x)) + \boldsymbol{b}_{hd}, \tag{10}$$

where $x$ denotes the input, $h_{CL}(\cdot)$ is the function represented by CLAdapter, $\phi_{pre}(\cdot)$ represents the pre-trained backbone, and $\boldsymbol{W}_{hd}$ and $\boldsymbol{b}_{hd}$ are the learnable parameters of the head. LP is computationally efficient as it only updates to the adapter and the head of the model. Although FT provides a thorough adaptation to the downstream task, starting CLAdapter training from scratch might distort the backbone's refined data representations, potentially exacerbating domain mismatch in OOD scenarios.

To address this, we propose a staged fine-tuning (SFT) strategy, beginning with only updates to the CLAdapter and heads in the first stage and progressing to full fine-tuning in the second. This method allows CLAdapter first to obtain better pre-transfer capabilities for the original pre-training domain, then further fine-tune the entire model to complete the downstream task. Since the fine-tuning overhead of LP is almost negligible compared to FT, the incremental cost of SFT is minimal. The SFT strategy can be represented as:

$$\mathcal{SFT}_{CL}(x) = \boldsymbol{W}_{hd}^{LP} \cdot h_{CL}^{LP}(\phi_{pre}(x)) + \boldsymbol{b}_{hd}^{LP}, \tag{11}$$

where $\boldsymbol{W}_{hd}^{LP}$, $\boldsymbol{b}_{hd}^{LP}$ and $h_{CL}^{LP}$ represent the head and CLAdapter after the first stage of fine-tuning (LP), respectively. Notably, in some cases, this LP stage often yields satisfactory performance for many tasks, thus reducing costs. SFT strategy efficiently leverages the strengths of both LP and FT, facilitating effective knowledge transfer from the pre-trained model to diverse downstream tasks.

# 4  Experiments

## 4.1  Experiment Setup

**Pre-training Dataset and Backbones.** In the era of big data, popular publicly available large-scale 2D datasets include the ImageNet-21K classification dataset [58] at the ten-million level, the LAION-400M image-text dataset [62] at the hundred-million level, and the LAION-2B image-text dataset [61] at the billion level. For the 3D domain, we utilize the Kinetics-400 [36], a large-scale dataset commonly used for action recognition, comprising approximately 260K video clips. On these large-scale datasets, we employ popular pre-trained models such as Vision Transformers (ViT) [70], ConvNeXt [50], and Video Swin Transformers (Swin) [49]. Details of these datasets and pre-trained backbones are listed in Table 1.

Table 1: Details of the pre-training datasets and the corresponding backbones used.

| Dataset | Scale | Type | Backbone |
|---|---|---|---|
| ImageNet-21K [58] | 14 Million | Images | ViT-B/16, ViT-L/16, ConvNeXt-B |
| LAION-400M [62] | 400 Million | Image-Text Pairs | ViT-B/16, ConvNeXt-B |
| LAION-2B [61] | 2000 Million | Image-Text Pairs | ViT-B/16, ConvNeXt-B |
| Kinetics-400 [8] | 0.26 Million | Video Clips | Swin-B, Swin-L |

**Downstream Tasks.** We experiment on 10 benchmarks across a broad spectrum of domains, including generic (Tiny-ImageNet [38]), multimedia (UCF101 [66] and HDMB51 [37]), industrial (InsPLAD-fault [78]), biological&medical (BreakHis [6] and HCRF [68]), agricultural (Apple Foliar Disease [72]), environmental&geographical (WHU-RS19 [81]), materials science (KTH-TIPS-2b [51]), OOD (PACS [40]), and 3D analysis (above video) to demonstrate the versatility and effectiveness of our CLAdapter. Full benchmark list (Table 5) and dataset descriptions with processing methods, are provided in *Appendix A.1*.

**Evaluation Metrics.** Following the previous works [78], we report the ROC on the InsPLAD-fault dataset. In alignment with established precedents [11], we utilize the precision, recall, accuracy, and F1 score as metrics on BreakHis and HCRF datasets. For all other datasets, we assess model efficacy using Top-1 accuracy. **Implementation Details** are included in *Appendix A.2*.

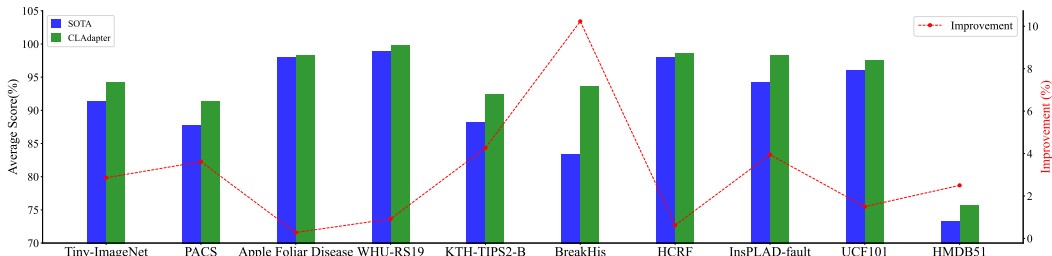

Figure 3: Performance comparison of CLAdapter against SOTA methods across various application domain datasets. The bar graph illustrates the average scores achieved by the CLAdapter and SOTA on each dataset, with the red fold line indicating the percentage improvement offered by the CLAdapter.

## 4.2  Main Results

**Overall Intuitive Performance Comparison.** Figure 3 provides an intuitive comparison between our method and prior SOTA approaches, while additional intuitive comparisons are presented in *Appendix B.1* Table 6. It demonstrates that our method consistently surpasses existing SOTA methods, highlighting its strong a in data-limited scientific domains and robustness against challenges such as limited data availability, domain distribution shifts, visual semantic variations, and fine-grained feature distinctions. More detailed quantitative comparisons are provided in the following sections.

**Comparison with SOTA approaches in diverse scientific domains.** As listed in Table 2, CLAdapter achieves state-of-the-art results in all benchmarks. Notably, on the industrial defect detection dataset InsPLAD-fault and the 3D multimedia action recognition UCF101 dataset, CLAdapter, fine-tuning

Table 2: Comparison of CLAdapter with SOTA methods for diverse scientific downstream tasks. Best performances are highlighted in **bold**, while the second-best are underlined.

(a) **Agricultural**:FoliarDisease [72]

| Method | Acc |
|---|---|
| MoCo v2 [13] | 96.04 |
| MaskCOV [69] | 95.82 |
| SPARE [87] | 96.70 |
| ViT [70] | 96.48 |
| DeiT [75] | 96.26 |
| TransFG [88] | 97.14 |
| Hybrid ViT [70] | 96.48 |
| Swin [48] | 98.08 |
| CLE-ViT [86] | 97.58 |
| **CLAdapter**ConvNeXt-B | **98.36** |
| **CLAdapter**ViT-B | **98.36** |

(b) **Geography**: WHU-RS19 [81]

| Method | Acc |
|---|---|
| DCA-Fusion [9] | 93.56 |
| GM [89] | 88.16 |
| GLM16 [89] | 92.99 |
| RSFJR [18] | 97.48 |
| MS2AP [7] | 98.88 |
| ViT [17] | 96.42 |
| Swin [48] | 97.12 |
| EMTCAL* [70] | 97.60 |
| SF-MSFormer [84] | 97.80 |
| **CLAdapter**ConvNeXt-B | **99.80** |
| **CLAdapter**ViT-B | **99.20** |

(c) **Materials**: KTH-TIPS2-B [51]

| Method | Acc |
|---|---|
| CDL [80] | 76.30 |
| Timofte [73] | 66.30 |
| DMD+IFV [52] | 76.20 |
| FV-VGGVD [14] | 88.20 |
| LETRIST [65] | 65.30 |
| CATex [21] | 66.70 |
| RAMBP [2] | 68.90 |
| TEX-Nets-LF [3] | 78.00 |
| BMCAnet [45] | 79.18 |
| **CLAdapter**ConvNeXt-B | **92.47** |
| **CLAdapter**ViT-B | **91.26** |

(d) **Biomedicine**: BreakHis [6]

| Method | Pre | Rec | Acc | F1 |
|---|---|---|---|---|
| ViT [17] | 80.02 | 80.73 | 84.89 | 80.37 |
| BotNet [67] | 79.20 | 80.72 | 85.32 | 79.50 |
| GasHis-Transformer [11] | 83.92 | 83.16 | 88.10 | 83.48 |
| LW-GasHis-Transformer [11] | 84.54 | 82.99 | 87.93 | 83.69 |
| **CLAdapter**ConvNeXt-B | **92.58** | **90.75** | **93.53** | **91.66** |
| **CLAdapter**ViT-B | **95.01** | **92.45** | **95.32** | **93.71** |

(e) **Biomedicine**: HCRF [68]

| Method | Pre | Rec | Acc | F1 |
|---|---|---|---|---|
| TransMed [15] | 94.34 | 97.06 | 95.58 | 95.58 |
| HCRF-AM [43] | 92.90 | 91.94 | 94.24 | 92.06 |
| GasHis-Transformer [11] | 98.55 | 97.38 | 97.97 | 97.97 |
| LW-GasHis-Transformer [11] | 95.99 | 96.90 | 96.43 | 96.43 |
| **CLAdapter**ConvNeXt-B | **98.61** | **98.57** | **98.57** | **98.59** |
| **CLAdapter**ViT-B | 95.01 | 95.00 | 95.00 | 95.00 |

(f) **Industrial**: InsPLAD-fault [78]

| Method | Glass Ins. | Light. RS. | Upper Sha. | Vari Grip | Yoke Sus. | Avg ROC |
|---|---|---|---|---|---|---|
| DifferNet [59] | 82.81 | 99.08 | 92.42 | 91.20 | 96.77 | 92.46 |
| AttentDifferNet [63] | 86.57 | 99.62 | 94.62 | 93.52 | 97.38 | 94.34 |
| FastFlow [1] | 70.16 | 82.02 | 77.43 | 65.54 | 71.48 | 73.33 |
| RD++ [63] | 86.21 | 97.54 | 83.67 | 93.85 | 92.46 | 90.75 |
| CS-Flow [55] | 85.73 | 96.60 | 88.40 | 91.53 | 90.70 | 90.59 |
| CFLOW-AD [22] | 82.22 | 95.52 | 86.60 | 90.37 | 83.87 | 87.72 |
| PatchCore [33] | 78.44 | 85.11 | 81.02 | 91.92 | 58.06 | 78.91 |
| **CLAdapter**ConvNeXt-B | **96.43** | **99.94** | **98.63** | **96.43** | **100.00** | **98.29** |
| **CLAdapter**ViT-B | **94.64** | **99.87** | **98.44** | **96.07** | **100.00** | **97.80** |

(g) **3D Multimedia**: Video Recognition

| Method | UCF101 [66] | HMDB51 [37] |
|---|---|---|
| MemDPC [23] | 86.10 | 54.50 |
| CoCLR [24] | 87.90 | 54.60 |
| RSPNet [12] | 93.70 | 64.70 |
| VideoMoCo [53] | 78.70 | 49.20 |
| Vi²CLR [16] | 89.10 | 55.70 |
| CVRL [54] | 94.40 | 70.60 |
| CORPf [29] | 93.50 | 68.00 |
| $\rho$BYOL$\rho = 4$ [20] | 94.20 | 72.10 |
| VideoMAE [74] | 96.10 | 73.30 |
| **CLAdapter**Swin-B | **97.60** | **75.80** |

uses only the first stage of SFT, achieving an average AUROC of 98.29% and an accuracy of 97.60%, respectively. This demonstrates the efficiency of CLAdapter in feature transformation and model fine-tuning across real-world scenarios. Moreover, CLAdapterConvNeXt-B and CLAdapterViT-B surpass the best-performing methods on the biomedical BreakHis dataset by 7.97% and 10.02% in F1 score, respectively, highlighting significant impact of CLAdapter on cross-domain medical with limited data. In summary, these results demonstrate the effectiveness of CLAdapter in leveraging knowledge from large-scale datasets to adapt and excel in various downstream tasks, pushing the boundaries of computer vision applications across different industry and science domains.

Table 3: Results on Tiny-ImageNet and PACS classic visual datasets. Architecture variants: ViT-L for Tiny-ImageNet, ViT-B for PACS. Best results are in **bold**, while the second-best are underlined.

| Tiny-ImageNet [38] | | PACS [40] (FT) | | PACS [40] (PEFT) | |
|---|---|---|---|---|---|
| Method | Acc | Method | Acc | Method | Acc |
| CaiT-S/36 [76] | 86.74 | Linear | 71.88 | VPT-Adapter[34] | 76.76 |
| DeiT-B/16-D [75] | 87.29 | Full | 87.79 | LoRA[28] | 88.53 |
| Swin-L/4 [48] | 91.35 | SFT | 88.91 | DoRA [46] | 88.43 |
| ViT-L [31] | 86.43 | L2-SP [26] | 87.74 | MoRA [35] | 89.09 |
| **CLAdapter**ViT-L | **94.21** | **CLAdapter** | **91.41** | **CLAdapter** | **91.41** |

**Comparison with Vision Models and Fine-Tuning Methods on ID and OOD Benchmarks.** As listed in Table 3, we evaluate CLAdapter on both general in-distribution (ID) and out-of-distribution (OOD) benchmarks to assess its effectiveness in adapting pre-trained models under data-limited condi-

tions. On the Tiny-ImageNet dataset, which serves as a general classification benchmark, CLAdapter substantially improves ViT-L's baseline performance from 86.43% to 94.21%, outperforming stronger architectures such as Swin-L by a margin of 2.86%. For cross-domain generalization, we conduct evaluations on the OOD benchmark PACS. CLAdapter achieves 91.41% accuracy, outperforming full fine-tuning (by 3.62%) and L2-SP (by 3.67%). When compared with parameter-efficient fine-tuning (PEFT) methods, CLAdapter surpasses VPT by a large margin of 14.65%, and maintains at least a 2.32% lead over recent approaches such as LoRA, DoRA, and MoRA. The results highlighting the robustness and efficiency of CLAdapter for both ID and OOD scenarios.

## 4.3 Ablation Study

**Discussion on Cluster Center Numbers.** The number of cluster centers $K$ in Equation (1) within the CLAdatper is an adjustable hyperparameter. An excessive number of cluster centers might cause the model to overfit, particularly in scenarios where downstream tasks offer limited numbers and diversity of samples. On the other hand, too few cluster centers may fail to transfer all data information, leading to underfitting. To explore an optimal value for this parameter, we conduct experiments on the BreakHis dataset using the ViT-B model pre-trained on LAION-2B. The results in Table 4 indicate that setting $K$ to 20 yields the most improvements for downstream tasks, with an optimal F1 score of 93.71% and an accuracy of 95.32%. Therefore, we recommend 20 as the default setting for the hyperparameter $K$.

Table 4: Comparison of fine-tuning results on the BreakHis dataset under different cluster center numbers $K$. The best results are in **bold**.

| Scores | Cluster Center Number ($K$) | | | | | | | | |
|---|---|---|---|---|---|---|---|---|---|
| | 5 | 10 | 15 | 20 | 25 | 30 | 100 | 200 | 300 |
| Acc | 92.81 | 92.81 | 92.45 | **95.32** | 94.24 | 93.88 | 93.52 | 91.73 | 92.81 |
| F1 | 90.40 | 90.73 | 90.41 | **93.71** | 92.59 | 92.14 | 91.77 | 89.03 | 90.42 |

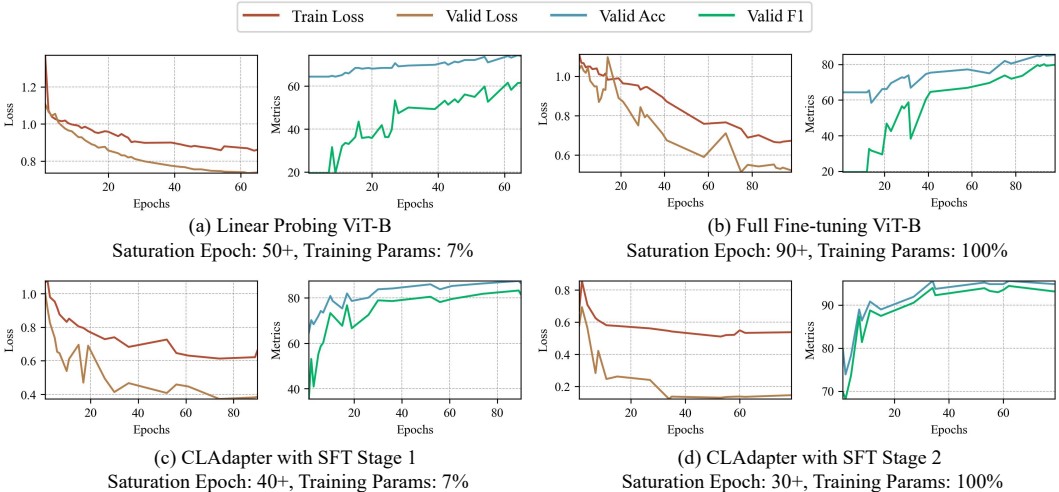

(a) Linear Probing ViT-B
Saturation Epoch: 50+, Training Params: 7%

(b) Full Fine-tuning ViT-B
Saturation Epoch: 90+, Training Params: 100%

(c) CLAdapter with SFT Stage 1
Saturation Epoch: 40+, Training Params: 7%

(d) CLAdapter with SFT Stage 2
Saturation Epoch: 30+, Training Params: 100%

Figure 4: Efficiency comparison of fine-tuning methods. CLAdapter achieves significant performance improvement with fewer training epochs and parameters, indicating both high effectiveness and efficiency. *Note that standard augmentations are only applied to the training set to mitigate overfitting but not to the validation set, which results in lower validation loss than training loss.*

**Analysis of Efficiency.** The proposed CLAdapter maintains high parameter efficiency while significantly improving performance. As detailed in *Appendix B.2* (Table 7), it introduces only 7–10.4% additional parameters to backbone models (ConvNeXt-B/ViT-B) with minimal computational overhead, yet achieves F1-score gains of up to 175.59%. Furthermore, CLAdapter also shows advantages in downstream task fine-tuning. As shown in Figure 4, by only fine-tuning 7% of the model parameters for 40 epochs in the first stage, CLAdapter achieves results comparable to full fine-tuning of

100% of the parameters for 90 epochs. Although the second stage of SFT requires tuning 100% of the parameters, saturation is reached in just 30 epochs, with an accuracy improvement of 12.44%. These experiments demonstrate the effectiveness and efficiency of our CLAdapter in fine-tuning and transferring pre-trained features.

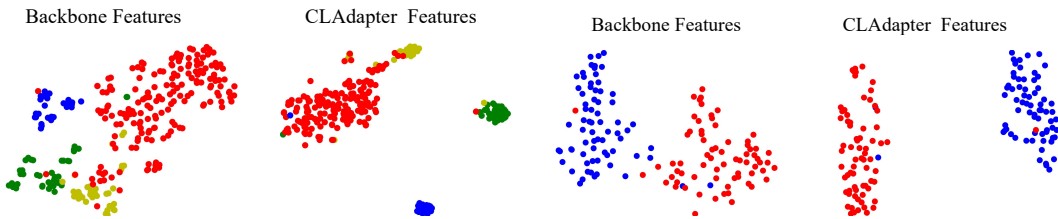

(a) BreakHis dataset: red for ductal, yellow for lobular, green for mucinous, and blue for papillary carcinoma.

(b) HCRF dataset: red for normal gastric slices and blue for cancerous gastric slices.

Figure 5: The $t$-SNE visualizations demonstrating class separability and compactness. The comparative analysis highlights the enhanced discriminability of features via CLAdapter.

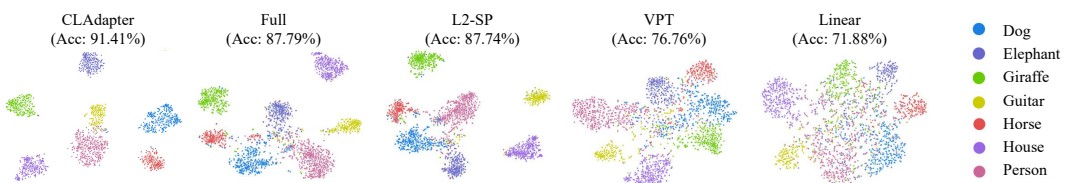

Figure 6: The $t$-SNE Feature visualization on PACS. The Top1-accuracy are reported additionally. CLAdapter demonstrated satisfactory separability and compactness.

**Visual Analysis of Features.** To further validate the effectiveness of CLAdapter in feature transfer, Figure 5 visualizes the features using t-distributed stochastic neighbor embedding ($t$-SNE) on the BreakHis and HCRF datasets. It compares the features extracted by the Backbone and those refined by CLAdapter. In the BreakHis dataset, post-CLAdapter application, the classes exhibit more distinct clustering. The ductal, lobular, mucinous, and papillary carcinomas are more separable and demonstrate increased intra-class compactness, underlining the robustness of CLAdapter in feature representation. Similarly, on the HCRF dataset, CLAdapter maximizes the inter-class distances of samples, effectively distinguishing between normal and cancerous gastric slices. In addition, to assess the quality of features extracted, we visualize the feature distributions for all fine-tuning methods on PACS test set using $t$-SNE. The experiments are performed based on the ViT-B model pre-trained on IageNet-22K. As shown in Figure 6, CLAdapter significantly improves the separability of representation vectors of different classes, exhibiting superior representational capacity.

## 5   Conclusion & Limitation

This work introduces the Cluster Attention Adapter (CLAdapter), a novel method designed to bridge the gap between large-scale pre-training on diverse datasets and fine-tuning for data-limited downstream tasks, particularly in diverse scientific domains. By leveraging attention mechanisms and clustering techniques, CLAdapter refines and adapts pre-trained models to enhance their performance significantly on a wide array of downstream tasks, showcasing superior adaptability and effectiveness. In addition, benefiting from our unified interface design, CLAdapter effortlessly merges with mainstream models. Moreover, an SFT strategy is presented to collaborate with CLAdapter to enhance the fine-tuning performance further. Through rigorous testing across ten diverse datasets, encompassing generic, multimedia, biological, medical, industrial, agricultural, environmental, geographical, materials science, OOD, and 3D analysis domains, CLAdapter all achieves a new state-of-the-art performance, highlighting its effectiveness in addressing the unique challenges of data scarcity and domain shift in scientific applications. **Limitations:** Currently, CLAdapter has not been specifically designed or validated for detection or segmentation. We leave these extensions for future work.

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

# A  Experiment Configuration

## A.1  Dataset Descriptions and Processing Methods

We experiment on 10 benchmarks across a broad spectrum of domains, including generic, multimedia, industrial, biological, medical, agricultural, environmental, geographical, materials science, OOD, and 3D analysis. This demonstrates the versatility and effectiveness of our CLAdapter. The datasets for each domain, class counts, and sample sizes are detailed in Table 5.

Table 5: Statistics of datasets used for evaluating downstream task performance.

| Dataset | Domains | Class | Train | Val | Test |
|---|---|---|---|---|---|
| Tiny-ImageNet [38] | General | 200 | 100000 | 10000 | 10000 |
| PACS [40] | General OOD | $4 \times 7$ | 1588 | 6355 | 2048 |
| BreakHis [6] | Biomedicine | 4 | 834 | 278 | 278 |
| HCRF [68] | Biomedicine | 2 | 70 | 70 | 140 |
| Apple Foliar Disease [72] | Agricultural | 4 | 1366 | - | 455 |
| WHU-RS19 [81] | Environmental Geography | 19 | 402 | 100 | 503 |
| KTH-TIPS-2b [51] | Materials Science | 11 | 3564 | - | 1188 |
| InsPLAD-fault [78] | Industrial | 5 | 5108 | - | 6417 |
| UCF101 [66] | 3D Multimedia | 101 | 9537 | - | 3783 |
| HMDB51 [37] | 3D Multimedia | 51 | 3570 | - | 1530 |

**Tiny-ImageNet.** Tiny-ImageNet [39] is a simplified version of the larger ImageNet dataset. It comprises 200 classes, each with 500 training images, 50 validation images, and 50 test images, resulting in a total of 100,000 images. Tiny-ImageNet serves as a benchmark for evaluating algorithms on a wide array of generic visual recognition tasks, testing both the depth and breadth of models' understanding of visual concepts. In the experiment, the data division adheres to official standards.

**PACS.** The PACS dataset [40] stands as a critical benchmark for assessing domain generalization capabilities in general computer vision. It contains images from four distinct domains: Photo, Art Painting, Cartoon, and Sketch, addressing a broad spectrum of visual styles and compositions. With seven common object classes across these domains, the dataset poses a significant challenge in learning domain-invariant features. It is particularly used for evaluating models on their ability to generalize from seen to unseen domains, making it an essential tool for research in domain adaptation and generalization. The Art Painting domain of the PACS dataset is exclusively utilized as the test set to assess cross-domain performance, while the remaining data are divided into training and validation sets in a 5-fold manner, with a ratio of 1:4.

**Apple Foliar Disease.** The Apple Foliar Disease dataset [71] is a specialized resource aimed at advancing the field of agricultural and plant disease recognition. It consists of high-quality images that capture various foliar diseases affecting apple leaves, including but not limited to apple scab, cedar apple rust, and powdery mildew, as well as images of healthy leaves for comparison. Leveraging such datasets not only validates our CLAdapter's cross-domain effectiveness in agriculture but also aids researchers and agronomists in enhancing precision agriculture, enabling timely and effective disease management to improve crop health and yield. The data split method follows the previous work as training and validation sets with a 3:1 ratio.

**WHU-RS19.** The WHU-RS19 dataset [82] is a high-resolution remote sensing dataset, primarily used for the evaluation of land cover and land use classification algorithms in the field of geographical and environmental analysis. Originating from the Wuhan University Remote Sensing Group, this dataset encompasses a diverse collection of 19 classes representing various natural and man-made features, including but not limited to agricultural lands, forests, water bodies, residential areas, and industrial sites. The images in WHU-RS19 are collected from different satellite and aerial sensors, challenging and enhancing classification models' robustness in geographical and environmental fields. For data partitioning, our experiments are consistent with previous studies [84].

**KTH-TIPS2-B.** The KTH-TIPS2-B dataset [51] is an extension of the KTH-TIPS dataset, both of which are designed for the task of texture classification and material recognition in the field of computer vision, particularly focusing on the challenges associated with variations in scale, pose, and illumination. This dataset is curated by the KTH Royal Institute of Technology in Sweden. KTH-TIPS2-B consists of images representing a set of 11 material categories, such as cotton, wool,

and aluminum, among others. Each material category includes images captured under different conditions and from multiple angles, providing comprehensive data for evaluating the performance of texture analysis algorithms. The data split method remains consistent with previous work [45].

**BreakHis.** The BreakHis dataset [6] comprises 7,909 breast cancer images across four magnification levels, divided into eight sub-classes. Originating from 82 anonymous patients in Brazil, BreakHis is a key dataset in digital breast histopathology research. Malignant tumor images at a $200\times$ magnification, including ductal carcinoma (DC), lobular carcinoma (LC), mucinous carcinoma (MC), and papillary carcinoma (PC), are used for classification. The dataset is split into training, validation, and testing sets in a 3:1:1 ratio, which is the same as the previous study [11].

**HCRF.** The HCRF dataset [68], well-known in gastric histopathology, consists of 560 cancerous and 140 normal images. Following the previous works [11], the HCRF dataset is divided into training, validation, and test sets as a 1:1:2 random stratified ratio. Available on Mendeley Data, it serves as an important resource for evaluating model performance in the field of computer vision and biomedicine.

**InsPLAD-fault.** The InsPLAD dataset [78], pivotal for advancing power line asset inspection, includes the InsPLAD-fault subset, a specialized collection designed for anomaly detection tasks in power line components. This subset harnesses real-world images captured by unmanned aerial vehicles (UAVs) of operational power line transmission towers, offering a unique challenge in the realm of industrial defect detection. It encapsulates five distinct categories of power line objects, facilitating deep learning models in effectively identifying and classifying anomalies. In the experiment, the data split method remains consistent with previous work [64].

**UCF101.** The UCF101 [66] is a widely recognized dataset in the field of action recognition, developed by the University of Central Florida. It's one of the most popular benchmarks for evaluating the performance of video-based action recognition algorithms. The dataset features 101 action categories, encompassing a broad range of activities such as sports, playing musical instruments, and human-object interactions. Each category in the UCF101 dataset consists of multiple video clips, amounting to over 13,000 clips and totalling more than 27 hours of video data. The videos are collected from YouTube and represent diverse actors, backgrounds, and lighting conditions. This diversity poses a challenge for action recognition systems, requiring the models to generalize across different environments and subject appearances. The data division method follows the official release.

**HMDB51.** The HMDB51 [37] is a comprehensive video dataset aimed at the task of human action recognition. Compiled by researchers from Brown University, it consists of 51 action categories, each containing at least 101 video clips, resulting in a total of over 6,800 clips. These actions span a wide array of human activities, including facial actions, general body movements, and interactions with objects. Due to the limited size of its dataset and the lack of diversity among samples, the HMDB51 dataset poses more challenges than UCF101. Additionally, the data splitting method is consistent with the official.

### A.2 Implementation Details

We meticulously design our experimental settings to ensure comparability and reproducibility.

For experiments on the Tiny-ImageNet, follow the protocols established in [32], ensuring consistency with prior benchmarks.

For 3D video analysis on UCF101 and HMDB51 datasets, our experimental configurations align with the previous methods [41], which facilitates direct comparison with existing SOTA approaches.

Regarding the remaining datasets, encompassing cross-domain and various real-world domain applications, we establish uniform implementation details to underscore the adaptability and convenience of CLAdapter. Specifically, we adopt an input resolution of $224 \times 224$ pixels across all experiments. The learning rate is initialized at $1e-4$, and models are trained for up to 100 epochs with a batch size of 16. We employ the AdamW optimizer, configuring it with momentum $\beta_1 = 0.9$ and a weight decay of $1e-3$, to adapt to the unique challenges presented by these varied datasets. The determination of cluster centers $K$ for CLAdapter is fixed at 20, balancing granularity and computational efficiency. Our experimental setup is powered by four Nvidia GeForce RTX 3090 GPUs, boasting 24 GB of memory, under the Ubuntu 20.04 environment. Python 3.8.3 is chosen as the programming language, with the PyTorch 1.13.1 framework being utilized for model development.

# B Additional Experimental Analyses

## B.1 Intuitive Performance Comparison of Our Method with SOTA and Baseline Methods

In Table 2 of the main manuscript, we present a comprehensive comparison of various methods across diverse scientific domains for downstream tasks (subtables a–g). Furthermore, Table 3 highlights the significant performance gains achieved by CLAdapter on both the in-distribution (ID) Tiny-ImageNet and the out-of-distribution (OOD) PACS benchmarks.

To provide a more intuitive understanding of the advantages of our method, we visualize the performance comparison, as shown in Figure 3. It can be seen that CLAdapter consistently outperforms state-of-the-art methods across different datasets. In addition, we listed all benchmark baseline results in Table 6. By using the best fine-tuning strategy and keeping the same backbone as ours for comprehensive evaluation, our method still maintains substantial improvements over both baselines and prior SOTA methods.

Table 6: Per-Domain Improvement over Baseline (ViT) and SOTA using CLAdapter.

| Improve%↑ | ID | OOD | Agricultural | Geography | Materials | Biomedicine | Industrial | 3D Multimedia |
|---|---|---|---|---|---|---|---|---|
| Ours *vs* Baseline | 7.8 | 3.6 | 2.0 | 2.8 | 2.5 | 12.2/13.6 | 4.6 | 2.5/4.1 |
| Ours *vs* SOTA | 2.9 | 3.6 | 0.3 | 0.9 | 4.3 | 10.0/0.6 | 4.0 | 1.5/2.5 |

## B.2 Analysis of Efficiency

As a universal adapter, our method exhibits relative efficiency under popular pre-trained models. Efficiency analysis results are listed in Table 7. For ConvNeXt-B and ViT-B models, CLAdapter adds only 10.4% and 7% more parameters, respectively, while slightly increasing computational complexity (Flops) by 0.44G for ConvNeXt-B and 1G for ViT-B. Despite this minimal increase in size and computation, CLAdapter achieves remarkable F1 score improvements of up to 175.59% for ConvNeXt-B and 51.46% for ViT-B.

Table 7: Comparison of Method efficiency.

| Method | Params(M) | Flops(G) | F1 | Train |
|---|---|---|---|---|
| ConvNeXt-B | 88.85 | 15.42 | 33.26 | 100% |
| **CLAdapter**$_{ConvNeXt-B-SFT-1}$ | 99.11 | 15.86 | 80.89 | 10.4% |
| **CLAdapter**$_{ConvNeXt-B-SFT-2}$ | 99.11 | 15.86 | 91.66 | 100% |
| ViT-B | 85.77 | 16.86 | 61.87 | 100% |
| **CLAdapter**$_{ViT-B-SFT-1}$ | 92.22 | 17.86 | 84.34 | 7.0% |
| **CLAdapter**$_{ViT-B-SFT-2}$ | 92.22 | 17.86 | 93.71 | 100% |

## B.3 Comparison of Using Different Scales of Pre-training Data

To delve into how the scale of large visual datasets influences cross-domain fine-tuning, we perform comparative experiments on the PACS and BreakHis datasets using ConvNeXt-B and ViT-B models pre-trained on three large-scale datasets, i.e., ImageNet-21K, LAION-400M, and LAION-2B.

Table 8: Results of using different pre-training resources on the PACS dataset.

| Method | ImageNet-21K | LAION-2B |
|---|---|---|
| Linear | 71.88 | 95.61 |
| Full | 87.79 | 47.17 |
| L2-SP | 87.74 | 45.56 |
| VPT | 76.76 | 97.46 |
| **CLAdapter** | **91.41** | **97.62** |

Table 8 lists the fine-tuning results on the PACS dataset based on ViT-B. It is observed that our CLAdapter achieves an accuracy of 97.62% under the LAION-2B pre-training dataset, which is enriched with more diverse knowledge, marking a 6.21% improvement over the smaller-scale ImageNet-21K dataset. Notably, both Full fine-tuning and L2-SP exhibit poor performance on LAION-2B, likely due to pattern collapse encountered during the transfer of massive pre-trained features, a problem that VPT and our CLAdapter circumvent through additional parameter transformation. Moreover, our CLAdapter's accuracy under ImageNet-21K pre-training surpasses that of VPT by 14.65%, indicating that CLAdapter is also capable of extracting and transforming a sufficient amount of downstream-relevant information from relatively smaller pre-training datasets.

Table 9: Results of using different pre-training resources on the BreakHis dataset.

| Method | ImageNet-21K | LAION-400M | LAION-2B |
|---|---|---|---|
| *SFT Stage 1* | | | |
| **CLAdapter**$_{\text{ConvNeXt-B}}$ | **80.89** | 71.72 | 75.17 |
| **CLAdapter**$_{\text{ViT-B}}$ | **84.34** | 80.63 | 80.67 |
| *SFT Stage 2* | | | |
| **CLAdapter**$_{\text{ConvNeXt-B}}$ | 88.60 | 90.55 | **91.66** |
| **CLAdapter**$_{\text{ViT-B}}$ | 90.19 | 91.35 | **93.71** |

The F1 score results on the BreakHis dataset are listed in Table 9. Whether combined with ConvNeXt-B or ViT-B pre-trained models, CLAdapter achieves the best fine-tuning results on larger pre-training datasets. Another notable observation is that when only the first stage of the SFT fine-tuning strategy is employed (i.e., freezing the weights of the pre-trained model), pre-training on the smaller-scale and knowledge-limited ImageNet-21K dataset yields better results. This suggests that freezing the pre-trained model weights limits CLAdapter's capacity for transformation, preventing the thorough transfer and refinement of rich knowledge to the downstream tasks. Nevertheless, even with just the first fine-tuning stage, CLAdapter still surpasses other SOTA methods on the BreakHis dataset (as listed in Table 2d).

