# OpenReview forum: "Unleashing Foundation Vision Models: Adaptive Transfer for Diverse Data-Limited Scientific Domains"
_NeurIPS.cc/2025/Conference — NeurIPS 2025 poster_

### Official Review · Reviewer_yjJZ · 2025-06-05

**Clarity:** 2
**Significance:** 2
**Originality:** 2
**Rating:** 5
**Confidence:** 3

**Summary:**

This paper introduces Cluster Attention Adapter (CLAdapter), a module designed to adapt large-scale pre-trained vision models like ViT and ConvNeXt to data-limited scientific domains. By leveraging attention mechanisms and cluster centers, CLAdapter personalizes feature transformation to enhance downstream task performance.

**Questions:**

1. Clarification on the Focus on Scientific Domains
  - CLAdapter seems like a general-purpose tuning method. Could the authors clarify why they emphasize scientific downstream tasks? Is there a technical reason (e.g., domain-specific distribution shifts) that makes CLAdapter particularly effective for these tasks, or were these domains selected due to better empirical results?
  - Suggestion: Please include experiments on more standard benchmarks (e.g., ImageNet, ImageNet-tiny is not enough) to demonstrate broader applicability beyond scientific datasets.
2. Comparison with Standard Tuning Methods
  - Since CLAdapter aims to compete with tuning methods like fine-tuning, linear probing, LoRA, and prompt tuning, it is unclear why the paper does not provide a controlled comparison under the same backbone architecture.
  - Suggestion: Please add direct comparisons of CLAdapter against these standard tuning methods using the same pre-trained model on the ten downstream tasks. This would help isolate the benefits of your method and strengthen the empirical evidence.
3. Sensitivity to the Number of Cluster Centers
  - The ablation study indicates that CLAdapter is quite sensitive to the number of cluster centers, which could affect its robustness across tasks.
  - Suggestion: Could you either propose an automatic way to set this hyperparameter or demonstrate that the performance does not degrade drastically over a reasonable range?
4. Figure 2 Visualization Improvement
  - The current version of Figure 2 is somewhat hard to interpret: trainable and frozen components are not clearly distinguishable, and the meaning of left-side annotations (e.g., N×D) is unclear.
  - Suggestion: Please revise the figure to more clearly distinguish between frozen and trainable parameters (e.g., using different shapes, labels, or highlighting) and clarify the meaning of each notation within the figure caption.
5. Clarification on Baselines in Table 1
  - Table 1 compares SOTA methods with different backbones, but it is unclear what tuning strategies were used for these baselines (e.g., full fine-tuning, linear probing).
  - Suggestion: Please explicitly state the tuning method used for each baseline in the table or in the experimental setup section.

**Ethical Concerns:**

["NO or VERY MINOR ethics concerns only"]

**Final Justification:**

My recommendation is to Accept.

I thank the authors for their detailed rebuttal. The responses have successfully addressed the main concerns and questions I raised in my initial review.

Resolved issues:  motivation of focus on scientific domains, visualization improvement

Unresolved issues: none

**Limitations:**

yes

**Quality:**

2

**Strengths And Weaknesses:**

## Strengths
- The visual analysis clearly demonstrates the difference in cluster distributions before and after applying CLAdapter, helping to intuitively validate the method’s effectiveness.
- The descriptions of the pretraining datasets and downstream tasks are detailed and comprehensive, providing a clear context for the experiments.

## Weaknesses
- CLAdapter appears to be a general-purpose tuning method rather than one specifically designed for AI4Science applications. It is unclear why the paper emphasizes scientific downstream tasks—does the method inherently favor these tasks, or were datasets selectively chosen to highlight favorable results? Moreover, the lack of experiments on standard datasets like ImageNet raises concerns about the method’s general applicability.
- Since CLAdapter targets transfer learning scenarios similar to fine-tuning, linear probing, LoRA, and prompt tuning, it is unclear why the main experiments do not include a direct comparison of these tuning methods under the same backbone. Instead, Table 1 compares different backbones, and it is not specified what tuning strategies were used for the SOTA baselines.
- The method is highly sensitive to the number of cluster centers in the ablation study, which raises concerns about its robustness and practical usability across different tasks and datasets.
- In Figure 2, some text elements are misaligned, and the visualization appears overly abstract. The color blocks make it hard to distinguish trainable and frozen parameters. Furthermore, while the left part annotates dimensions like N×D, these notations are not reflected or explained in the right part of the figure, leading to confusion about their intended meaning.

---

> ### Author Rebuttal · Authors · 2025-07-31
>
> # Response to Reviewer yjJZ.
>
> Thank you very much for recognizing the strengths of our work, including the effectiveness, clear background and description.
>
>
> Below, we address your constructive comments and suggestions, and describe the corresponding revisions we have made.
>
>
>
> ### Weaknesses 1 and Question 1:
> > 1. Clarification on the Focus on Scientific Domains
>
> >  - CLAdapter seems like a general-purpose tuning method. Could the authors clarify why they emphasize scientific downstream tasks? Is there a technical reason (e.g., domain-specific distribution shifts) that makes CLAdapter particularly effective for these tasks, or were these domains selected due to better empirical results?
> > - Suggestion: Please include experiments on more standard benchmarks (e.g., ImageNet, ImageNet-tiny is not enough) to demonstrate broader applicability beyond scientific datasets.
>
> ### **Authors’ Response:**
> Thank you for your comment.
> We would like to **clarify your misunderstanding** that:
> - **CLAdapter Design Motivation and Technical Reasons**
>     - CLAdapter is tailored for **data-limited scientific domin** rather than the **general-purpose tuning method**
>         - where **data-limited**, **domain distribution shifts**, **vision semantic variations**, **fine-grained feature discrimination.**
>         - This create unique challenges that differ from general-purpose tuning method.
>
> - **Scientific domins data selection principles**
>     - Each dataset selected for each scientific domin:
>         - highly cited
>         - popular
>         - representative
>     - They are **not** ***"chosen to highlight favorable results"***.
>         - Otherwise, it would be impossible to cover more than 10 downstream scenarios.
>
> - **Regarding the use of ImageNet** (***This is a pre-train benchmark, not transfer fine-tuning.***)
>     - Standard/large benchmarks ImageNet are completely **outside the scope of our research.**
>     - **In contrast**, our research aims to leverage models **pre-trained on a general large-scale dataset** (e.g., ImageNet-1K/22K, LAION-400M/2B, etc) to **fine-tune cross-domin transfer** for data-limited **scientific** **domains**.
>
>
> ### Weaknesses 2 and Question 2:
> > Since CLAdapter targets transfer learning scenarios similar to fine-tuning, linear probing, LoRA, and prompt tuning, it is unclear why the main experiments do not include a direct comparison of these tuning methods under the same backbone. Instead, Table 1 compares different backbones, and it is not specified what tuning strategies were used for the SOTA baselines.
>
> > 2. Comparison with Standard Tuning Methods
> > - Suggestion: Please add direct comparisons of CLAdapter against these standard tuning methods using the same pre-trained model on the ten downstream tasks. This would help isolate the benefits of your method and strengthen the empirical evidence.
>
> ### **Authors’ Response:**
> Thank you for your comment.
> We would like to clarify that:
>
> - **Experimental Comparison Motivation**:
>      - Our goal is to demonstrate that **CLAdapter achieves SOTA performance across various data-limited scientific domains**, using only **mainstream backbones (ViT, etc)**.
>      - Thus, we primarily compare with **recent publication SOTA methods  on each dataset domin** (Specifically designed for this downstream task with **standard tuning**).
>
>
> - **Following your valuable suggestion**, we compared the performance of the **same** pre-trained **model** (ViT, standard tuning) with CLAdapter **on ten downstream tasks**.
>     - | Improve%↑       | ID  | OOD | Agricultural | Geography | Materials | Biomedicine | Industrial | 3D Multimedia |
>       |-----------------|-----|-----|--------------|-----------|-----------|-------------|------------|---------------|
>       | CLAdapter ***VS*** Same ViT Baseline(pre-trained with ImageNet) | +7.8 | +3.6 | +2.0          | +2.8       | +2.5       | +12.2, +13.6   | +4.6        | +2.5, +4.1       |
>
> This comparison further vividly highlights the strengthened benefits of our method.
>
>
>
>
> ### Weaknesses 3 and Question 3:
> > 3. Sensitivity to the Number of Cluster Centers
> > - The ablation study indicates that CLAdapter is quite sensitive to the number of cluster centers, which could affect its robustness across tasks.
> > - Suggestion: Could you either propose an automatic way to set this hyperparameter or demonstrate that the performance does not degrade drastically over a reasonable range?
>
>
> ### **Authors’ Response:**
> Thank you for your comment.
> We would like to clarify that:
>
> - CLAdapter is **not sensitive** when the number of cluster centers is set within a **reasonable range (20–50)**.
>     - | Num. of Clusters  | 20   | 25   | 30   | 35   | 40   | 45   | 50   |
>       |-------------------|------|------|------|------|------|------|------|
>       | Accuracy (%)      | 95.32| 94.24| 93.88| 93.98| 94.01| 93.85| 93.82|
>       | F1 Score (%)      | 93.71| 92.59| 92.14| 92.84| 92.11| 92.82| 92.77|
>
>
> - The results shown in the paper include **extreme values (e.g., 100，200， even 300)** to stress-test the model’s efficiency in very high number cluster centers settings, **which are not needed in practice.**
>     - This may have **led to your misunderstanding**, and we will explain it in the paper.
>
>
> - Possibility of *"**Automating** this parameter setting"*
>     - using a classic **simple grid search** within our recommend range (20-50).
>     - In addition, in future work, we also plan to explore smarter automation methods.
>
>
> ### Weaknesses 4 and Question 4:
> > 4. Figure 2 Visualization Improvement
> > - The current version of Figure 2 is somewhat hard to interpret: trainable and frozen components are not clearly distinguishable, and the meaning of left-side annotations (e.g., N×D) is unclear.
> > - Suggestion: Please revise the figure to more clearly distinguish between frozen and trainable parameters (e.g., using different shapes, labels, or highlighting) and clarify the meaning of each notation within the figure caption.
>
> ### **Authors’ Response:**
> Thank you for your careful observation.
>
> - We have revised **Figure 2** to **explicitly distinguish** frozen and trainable modules using light blue and orange components, and **explicitly** added in the labeled below the diagram for better readability
> - The notations have also been clarified:
>   - $N$: number of patch tokens
>   - $D$: latent space dimension (feature embedding dimension)
>
> We appreciate your feedback on improving the clarity of our visualizations.
>
>
> ### Question 5:
> > 5. Clarification on Baselines in Table 1
> > - Table 1 compares SOTA methods with different backbones, but it is unclear what tuning strategies were used for these baselines (e.g., full fine-tuning, linear probing).
> > - Suggestion: Please explicitly state the tuning method used for each baseline in the table or in the experimental setup section.
>
> ### **Authors’ Response:**
> Thank you for your comment.
> - These SOTA methods all follow the standard full fine-tuning specified in their papers, and we will add more information in Experiment Configuration Section.
>
> We sincerely hope that these improvements can facilitate higher scores for our work.

---

> > ### Comment · Reviewer_yjJZ · 2025-08-01
> > **Subject: Response to Rebuttal**
> >
> > Thank you for your detailed rebuttal, which has resolved many of my concerns.
> >
> > Regarding Weakness 2 and Question 2, I initially felt my question was not directly answered. I will restate it here for clarity:
> >
> > To demonstrate the effectiveness of CLAdapter and its advantage of having fewer tunable parameters, it seems the most controlled experiment would be to isolate the tuning method as the sole variable. This would require keeping the backbone, fine-tuning data, and test data consistent while comparing the performance of CLAdapter against other parameter-efficient methods like linear probing, LoRA, and prompt tuning. Therefore, the comparison against full fine-tuning alone is not entirely conclusive, as full fine-tuning can be prone to overfitting in the scenarios you mentioned (e.g., data-limited settings, domain distribution shifts, etc.).
> >
> > However, I have since reviewed your discussion with Reviewer AwdT, and I found the relevant analysis and comparison in that response. This has fully addressed my question.
> >
> > Therefore, I have no further concerns. Thank you.

---

> > > ### Author Response · Authors · 2025-08-01
> > > **Acknowledgment of Reviewer’s Feedback and Appreciation**
> > >
> > > We sincerely appreciate your positive and timely feedback, and we are pleased to hear that our response has resolved your concerns. Thank you again for your thoughtful review !

---

### Official Review · Reviewer_AwdT · 2025-06-30

**Clarity:** 3
**Significance:** 4
**Originality:** 3
**Rating:** 4
**Confidence:** 4

**Summary:**

This paper proposes a novel method called Cluster Attention Adapter (CLAdapter), which addresses the challenges of transferring large-scale pre-trained vision models to data-limited downstream tasks in scientific domains by leveraging feature clustering and a progressive training strategy.

**Questions:**

1. In typical classification tasks using Vision Transformers (ViTs), only the class token is used for classification. However, this paper discards the class token and instead adopts a more complex approach based on patch token aggregation. Is there any experimental evidence supporting this design choice? Additionally, why does the class token have a negative effect in the experimental scenarios considered in this work? In Table 1, do all experiments based on ViT models use the same type of output token for classification?

  2. In Figure 3, are the training loss and validation loss labels possibly reversed?

  I would consider increasing my score if the authors address the identified weaknesses and provide satisfactory responses to the questions raised in the review.

**Ethical Concerns:**

["NO or VERY MINOR ethics concerns only"]

**Final Justification:**

The author has effectively addressed my concerns and further demonstrated the efficacy of CLAdapter with more comprehensive experiments.

**Limitations:**

The paper mentions that CLAdapter is not applicable to detection and segmentation tasks. However, as a discriminative task, semantic segmentation is not fundamentally different from image classification. It may be worth exploring the application of CLAdapter to semantic segmentation tasks, although the computational cost could be higher.

**Quality:**

3

**Strengths And Weaknesses:**

- strength
  1. By incorporating an attention mechanism and learnable clustering centers, CLAdapter adaptively refines the feature representations of pre-trained models, resulting in significantly improved transfer performance in data-limited scientific domains.

  2. The proposed two-stage fine-tuning strategy first updates only the adapter and classification head, and then performs full model fine-tuning. This approach effectively balances computational efficiency with performance optimization.

- weaknesses
  1. While a standard classification head typically uses a single linear layer (i.e., linear probing), CLAdapter introduces a new form of classification head that includes nonlinear modules and a larger number of parameters. This makes the experimental comparison potentially unfair. It is strongly recommended that the authors demonstrate that CLAdapter still achieves significantly better performance when compared to nonlinear classification heads with a comparable number of parameters.

  2. Similarly, for the SFT setting, it is recommended to replace CLAdapter with a simple nonlinear classification head that has a comparable number of parameters, and then perform a comparison using the same SFT strategy.

  3. The experimental comparisons appear somewhat disorganized. To properly demonstrate the effectiveness of CLAdapter, a consistent set of pre-trained models (e.g., {MAE-B, MAE-L, DINO-B, DINOv2-B, ...}) should be used across different datasets, comparing their performance with and without CLAdapter. However, in the current version of the paper, even the baseline results for ConvNeXt are missing, which makes it difficult to assess the actual improvement brought by CLAdapter.

  4. Recently, self-supervised pre-trained models have become widely adopted. However, the paper lacks experiments using recent self-supervised models such as DINO, DINOv2, and the CLIP family as base models. This omission makes it unclear whether CLAdapter can provide benefits when applied to modern, high-performing pre-trained models.

---

> ### Author Rebuttal · Authors · 2025-07-31
>
> # Response to Reviewer AwdT on more comparisons and details questions.
>
> Thank you for recognizing CLAdapter and SFT's novel design, efficiency, and value for data-limited scientific domains.
>
> We sincerely appreciate your openness to reconsider the score upon addressing the concerns. We’ll carefully revise to fully respond to your insightful feedback.
>
>
> ### Question 1:
> > In typical classification tasks using Vision Transformers (ViTs), only the class token is used for classification. However, this paper discards the class token and instead adopts a more complex approach based on patch token aggregation. Is there any experimental evidence supporting this design choice? Additionally, why does the class token have a negative effect in the experimental scenarios considered in this work? In Table 1, do all experiments based on ViT models use the same type of output token for classification?
>
>
> ### **Authors’ Response:**
>
> Thank you for the question.
>
> - **Training from Scratch (without loading pre-trained weights)**
>     - use class token
>         - [cls_token] -> classification
>     - not use class token
>         - patch token aggregation (features embedding, via Global Average Pooling)
>     - **Both** yield almost **similar performance**.
>         - **This has been proven** by theoretical analysis and empirical results **in [1] (refer to "ViT paper Figure 9")** (*ICLR 2021*).
>
>     - Therefore, in this setting, the use of the **class token is indifferent**.
>
> - **Fine-Tuning (focus in this paper)**
>     - use **class token often harms performance**
>         - ***especially in cross-domain transfer to data-limited scientific domin***
>         - Because the class token tends to **preserve the pre-training domain-specific** deep priors.
>         - **Theoretical and experimental** **proofs**, please **refer [2]** (*NeurIPS 2024*)
>
>
> Moreover, influential works like **MAE** [3] and the **BEiT series** [4] also **discarded the class token during downstream classification**.
>
>
>
> - Given this, our CLAdapter intentionally discards the class token and instead focuses on **mining transferable patterns from the upstream patch embeddings**, which better supports adaptation to data-limited scientific domains.
>
> - In addition, regarding Table 1, we confirm that all ViT models use the same type of output token for classification (The experiments are based on timm framework).
>
> ___
>
> ### Question 2:
> > In Figure 3, are the training loss and validation loss labels possibly reversed?
>
>
> ### **Authors’ Response:**
> Thank you for your careful observation — we appreciate your attention to detail.
>
> - We understand your concern may come from the fact that **the validation loss appears lower than the training loss in Figure 3**,
>     - which **leads you to think** ***"loss labels possibly reversed?"***
>
> - However, the labels in Figure 3 are **correct**.
>     - This is because our training pipeline applies **standard data augmentations** (e.g., random rotation, flip, brightness/contrast adjustment) **only to the training set**, but **not to the validation set**.
>     - This results in the training loss being higher than the validation loss.
>
>
> ___
>
> ### Weaknesses 1:
> > While a standard classification head typically uses a single linear layer (i.e., linear probing), CLAdapter introduces a new form of classification head that includes nonlinear modules and a larger number of parameters. This makes the experimental comparison potentially unfair. It is strongly recommended that the authors demonstrate that CLAdapter still achieves significantly better performance when compared to nonlinear classification heads with a comparable number of parameters.
>
> ### **Authors’ Response:**
> Thank you for raising this comment.
> - We would like to clarify that:
>     - **CLAdapter is an adapter module inserted between the backbone and the classification head**,
>     - and is ***"not a new form of classification head"*** itself.
>     - In addition, in our CLAdapter-related image experiments, the **classification head remains a standard single linear layer**, consistent with conventional practice.
>     - Therefore, comparing CLAdapter to more complex classification heads is not the focus.
>
> - However, **to address your concerns**, we still provide the following comparison:
>     - **Nonlinear head** (params-similar) set:
>         - Image filed is mostly done with linear heads, but some video field research introduce Spatio-Temporal Attention Heads (nonlinear) [5].
>     - Results on HMDB51 dataset with Pre-trained Swin Backbone
>         - | Method | Classification Head Type | Accuracy (%) |
>           |--------|---------------------------|---------------|
>           | Baseline | Linear Head | 71.70 |
>           | Nonlinear Head | Spatio-TemporalAttention3D Head| 72.11 |
>           | CLAdapter | Linear Head | **75.80** |
>     - This comparison highlight the superior performance of CLAdapter.
>
>
>
>
> ___
>
> ### Weaknesses 2:
> > Similarly, for the SFT setting, it is recommended to replace CLAdapter with a simple nonlinear classification head that has a comparable number of parameters, and then perform a comparison using the same SFT strategy.
>
> ### **Authors’ Response:**
> - Similarly, the clarifications we need to make are consistent with the response to weakness 1.
>
> - Meanwhile, we also added a corresponding SFT strategy for above Table Nonlinear Head (Spatio-TemporalAttention3D):
>     - Nonlinear Head + SFT = 72.98 (**+ 0.87%**)
>
> This shows that the **SFT** strategy has a slight **gain for the nonlinear head**, but it is **still far behind our CLAdapter**.
>
>
> ___
>
> ### Weaknesses 3:
> > The experimental comparisons appear somewhat disorganized. To properly demonstrate the effectiveness of CLAdapter, a consistent set of pre-trained models (e.g., {MAE-B, MAE-L, DINO-B, DINOv2-B, ...}) should be used across different datasets, comparing their performance with and without CLAdapter. However, in the current version of the paper, even the baseline results for ConvNeXt are missing, which makes it difficult to assess the actual improvement brought by CLAdapter.
>
> ### **Authors’ Response:**
> Thank you for your suggestion.
>
> - We would like to clarify two key points:
>     - (1) **Experimental Comparison Motivation**:
>        - Our goal is to demonstrate that **CLAdapter achieves SOTA performance across various data-limited scientific domains**, using only **mainstream backbones (ViT, ConvNeXt)**.
>        - Thus, we primarily compare with **recent publication SOTA methods on each dataset domin** rather than exhaustively combining all pre-trained models.
>
>     - (2) **Ablation study w/o CLAdapter**
>         - We mainly conducted ablation (***w/o  CLAdapter***) based **ViT** for **all** datasets (***Appendix Table 6***).
>         - In addition, ConvNeXt, CLIP only on **part** dataset (***Appendix Table 9***)
>
> - **Following your valued suggestions**, we now supplement the corresponding ablation results to demonstrate the **further improvement of CLAdapter**.
>
>     - | Method | FoliarDisease | HCRF | HMDB51|
>       |--------|---------------------------|---------------|----|
>       | MAE-B    | 97.35 | 92.11 | 73.30
>       | +CLAdapter | 97.98 | 92.58 | 74.81
>       | MAE-L    | 96.21 | 93.44 | -|
>       | +CLAdapter  | 97.98 | 96.58 | -|
>       | DINO-B  | 96.12 | 95.94 |59.95|
>       | +CLAdpater  | 98.24 | 97.22| 62.16|
>       | DINOv2-B  | 97.41 | 96.98 | 60.21|
>       | +CLAdapter  | 98.68 | 98.59 | 63.34|
>       | ConvNeXt-B | 96.48| 95.43 |  58.6|
>       | +CLAdpter | 98.36| 98.59 | 62.99|
>
>
>
> ---
>
> ### Weaknesses 4:
> > Recently, self-supervised pre-trained models have become widely adopted. However, the paper lacks experiments using recent self-supervised models such as DINO, DINOv2, and the CLIP family as base models. This omission makes it unclear whether CLAdapter can provide benefits when applied to modern, high-performing pre-trained models.
>
> ### **Authors’ Response:**
>
> - We would like to **clarify** that **we have included CLIP series** in manuscript (***Appendix Table 9***).
>     - We consider not only the high performance and impact of CLIP, but also its versions with different pre-training data sizes (LAION-400M and LAION-2B).
>     - But so soory, we **only wrote the CLIP versions *“LAION-400M” and “LAION-2B”***, which ****caused your misunderstanding and neglect****
>
> - We now write its name **more clearer as follows**:
>     - The results on the BreakHis show that **CLAdapter can be applied to** modern, high-performing pre-trained models, and large-scale pre-training data.
>     - | Method | CLIP-LAION-400M | CLIP-LAION-2B |
>       |--------|---------------------------|---------------|
>       | $\textbf{CLAdapter}_{\text{ConvNeXt-B}}$    | 90.55 | 91.66 | 73.30
>       | $\textbf{CLAdapter}_{\text{ViT-B}}$ | 91.35 | 93.77 |
>
> - For the DINO series, the previous response (Weaknesses 3) also demonstrated.
>
>
> ---
>
> ### Reference:
> [1] Dosovitskiy, Alexey, et al. "An image is worth 16x16 words: Transformers for image recognition at scale." arXiv preprint arXiv:2010.11929 (2020).
>
> [2] Zou, Yixiong, et al. "A closer look at the CLS token for cross-domain few-shot learning." Advances in Neural Information Processing Systems 37 (2024): 85523-85545.
>
> [3] He, Kaiming, et al. "Masked autoencoders are scalable vision learners." Proceedings of the IEEE/CVF conference on computer vision and pattern recognition. 2022.
>
> [4] Bao, Hangbo, et al. "BEIT: Bert pre-training of image transformers." arXiv preprint arXiv:2106.08254 (2021).
>
> [5] Pandey, et al. "Resstanet: deep residual spatio-temporal attention network for violent action recognition." International Journal of Information Technology 16.5 (2024): 2891-2900.

---

> > ### Comment · Reviewer_AwdT · 2025-08-08
> >
> > I appreciate the authors' valuable response. As stated before, I have accordingly raised my score to *Borderline Accept*.

---

### Official Review · Reviewer_fSgx · 2025-07-03

**Clarity:** 3
**Significance:** 4
**Originality:** 4
**Rating:** 5
**Confidence:** 4

**Summary:**

Data scarcity remains a significant challenge in scientific vision tasks, limiting the effective training of vision models. While existing transfer learning frameworks perform well when downstream tasks are closely aligned with the distribution of the pretraining data, they often fall short in scientific applications where data is both limited and typically out-of-distribution.
To address this, the paper proposes CLAdapter combined with a Staged Fine-Tuning strategy, which adapts pretrained representations—whether from ViT-based or CNN-based models—to better suit downstream tasks under low-data, out-of-distribution conditions.

**Questions:**

It would be valuable to more clearly highlight how CLAdapter improves performance relative to existing methods, particularly in the context of varying downstream dataset sizes. Specifically, explicitly reporting the number of training examples available for each downstream task would help contextualize the challenges of low-data regimes. Emphasizing CLAdapter’s robustness in tasks with extremely limited data would strengthen the case for its effectiveness, especially in out-of-distribution or scientific vision scenarios where data scarcity is common. This comparison would also underscore the practical advantages of the proposed approach in real-world settings where large-scale annotated datasets are unavailable.

**Ethical Concerns:**

["NO or VERY MINOR ethics concerns only"]

**Final Justification:**

In rebuttal, the addition of a detailed comparison table showing performance gains across a diverse set of scientific and general datasets with corresponding train sizes effectively demonstrates the method’s robustness in low-data and out-of-distribution settings.
This added analysis further supports the practical relevance of CLAdapter, especially in real-world scenarios where annotated data is limited.
Given the originality of the proposed approach, the strong experimental results, the other reviewers' comments, I maintain my rating.

**Limitations:**

yes

**Paper Formatting Concerns:**

No concerns.

**Quality:**

3

**Strengths And Weaknesses:**

- The idea of using learnable cluster centroids to model the features relevant to the downstream task is novel and interesting.
- The paper is technically sound. Extensive experiments and results support the claims.
- The results are impactful for different scientific domains and the generalizability makes it even more valuable.
- Highlighting how CLAdapter boosts the performance as compared to the existing methods, while taking into consideration the downstream task’s dataset size would be beneficial.

---

> ### Author Rebuttal · Authors · 2025-07-31
>
> # Response to Reviewer fSgx on more clearly highlighting CLAdapter's value.
>
> Thank you very much for recognizing the strengths of our work, including the novel and interesting design, the strong empirical validation, and the impactful results and valuable across general and scientific domains.
>
>
> Below, we address your constructive comments and suggestions, and describe the corresponding revisions we have made.
>
>
>
> ### Comment 1:
> > It would be valuable to more clearly highlight how CLAdapter improves performance relative to existing methods, particularly in the context of varying downstream dataset sizes. Specifically, explicitly reporting the number of training examples available for each downstream task would help contextualize the challenges of low-data regimes. Emphasizing CLAdapter’s robustness in tasks with extremely limited data would strengthen the case for its effectiveness, especially in out-of-distribution or scientific vision scenarios where data scarcity is common. This comparison would also underscore the practical advantages of the proposed approach in real-world settings where large-scale annotated datasets are unavailable.
>
> ### **Authors’ Response:**
> Thank you for this important suggestion.
>
> The aspects you highlighted: ***training sample size*** &  ***performance gains*** & ***effectiveness under limited-data*** & ***scientific scenarios*** are indeed central to the **core value of CLAdapter**.
>
> Presenting these **clearly helps underscore** CLAdpater practical advantages in real-world settings.
>
> - **Following your valuable suggestion**, we compared these dimensions in the following unified table:
>
>     | **Dataset (Domain)** | **Train Size** | **CLAdapter vs. Baseline (\%)** | **CLAdapter vs. SOTA (\%)** |
>     |----------------------|----------------|----------------------------------|------------------------------|
>     | Tiny-ImageNet (General)         | 100,000     | +7.8                          | +2.9                        |
>     | PACS (General OOD)              | 1,588       | +3.6                          | +3.6                        |
>     | Apple Foliar Disease (Agri)     | 1,366       | +2.0                          | +0.3                        |
>     | WHU-RS19 (Geography)            | 402         | +2.8                          | +0.9                        |
>     | KTH-TIPS2-B (Materials)         | 3,564       | +2.5                          | +4.3                        |
>     | **BreakHis (Biomedicine)**      | **834**     | **+12.2**                     | **+10.0**                   |
>     | **HCRF (Biomedicine)**          | **70**      | **+13.6**                     | **+0.6**                    |
>     | InsPLAD-fault (Industrial)      | 5,108       | +4.6                          | +4.0                        |
>     | UCF101 (3D Multimedia)          | 9,537       | +2.5                          | +1.5                        |
>     | HMDB51 (3D Multimedia)          | 3,570       | +4.1                          | +2.5                        |
>
>
> - **In addition** to **highlight** the above revised content in the **paper body text**, we have **also included a more clearly and comprehensive presentation** in:
>     -  Appendix A.2 (Table 5, dataset details)
>     -  Appendix B.1
>         -  Figure 5, clear & vivid comparison **bar chart and line plot**
>         -  Table 6, clear & detail comparison

---

> ### Comment · Reviewer_fSgx · 2025-08-06
>
> Thank you for your detailed and thoughtful rebuttal.
>
> I appreciate the efforts made to incorporate the suggestions regarding training sample size, performance gains, and the model's effectiveness under limited-data and scientific settings. I'm satisfied with the corresponding updates in the appendix and the way feedback has been addressed.

---

> > ### Author Response · Authors · 2025-08-07
> > **Acknowledgment of Reviewer’s Feedback and Appreciation**
> >
> > We sincerely appreciate your positive and timely response.

---

### Official Review · Reviewer_UMqW · 2025-07-19

**Clarity:** 3
**Significance:** 4
**Originality:** 3
**Rating:** 5
**Confidence:** 3

**Summary:**

This paper introduces CLA dapter, a cluster-based attention adapter designed for efficient adaptation of foundation vision models to data-limited scientific domains. By leveraging attention mechanisms and distribution-aware cluster centers, CLAdapter refines pretrained features to better match diverse downstream tasks. It supports both CNNs and Transformers in 2D/3D settings through a unified interface. Experimental results show promising performance, demonstrating the effectiveness in enabling adaptive transfer. The method also achieved top rankings in multiple scientific vision challenges, further validating its practical effectiveness in adaptive transfer.

**Questions:**

1. Please provide more analysis/discussion with Lora based methods.
2. Please provide more analysis on inference efficiency or adapter compression.

**Ethical Concerns:**

["NO or VERY MINOR ethics concerns only"]

**Limitations:**

Yes.

**Paper Formatting Concerns:**

No.

**Quality:**

4

**Strengths And Weaknesses:**

Strengths:
1. The CLA adapter design is novel. It introduces multiple residual adapters inserted into multiple layers
2. The method is validated on multiple datasets covering a wide range of topics, demonstrating versatile transferability in both classification and segmentation tasks.
3. The method ranks top on multiple vision-related competitions, which reinforces its real-world effectiveness

Weaknesses:
1. Please provide more analysis/discussion with Lora based methods.
2. Please provide more analysis on inference efficiency or adapter compression
3. Please add the dataset/paper reference (e.g. PACS) in the paper body text instead of adding them in the table.

---

> ### Author Rebuttal · Authors · 2025-07-31
>
> # Response to Reviewer UMqW on more Lora-based methods analysis, more inference efficiency analysis, and dataset description position.
>
> Thank you very much for your recognition of the good performance, novel design, extensive validation, and real-world effectiveness.
>
> Your constructive comments and suggestions are exceedingly helpful to improve our paper. We have carefully incorporated them in the revised paper. In the following, your comments are first stated and then followed by our point-by-point responses.
>
> ### Comment 1:
> > Please provide more analysis/discussion with LoRA-based methods.
>
> ### **Authors’ Response:**
>
> Thank you for your suggestion. Our **CLAdapter differs significantly from LoRA-based methods** in both *design focus* and *target domains*.
>
> - **1. Design & Applicability Analysis/Discussion**
>     - | Aspect | **LoRA-based** | **CLAdapter (Ours)** |
>       |--------|------------------------|---------------------- |
>       | **Applicable Backbone** | Transformer-based | CNN-based & Transformers-based & their 3D versions |
>       | **Design Motivation** | Large scale **LLM** Efficient training | Cross-domain Transfering: ***foundation vision*** models → **data-limited scientific domains**. |
>       | **Application Domain** | General tasks | *Scientific & Data-Limited domains*: medical, biological, materials, environmental, etc. |
>
>
>
> - **2. Performance Analysis/Discussion**
>     - We already included a ViT-L backbone in the manuscript (Table 2) to provide a fair comparison with LoRA. On **PACS** (popular datasets in the AI field), CLAdapter achieves **91.41%**, surpassing LoRA (**88.53%**) by **+2.88%**. We also outperform advanced versions (**DoRA** and **MoRA** by **+2.98%** and **+2.32%**, respectively).
>
>     - If the **comparison** domain is **changed** to the ***data-limited scientific domain*** where **CLAdapter excels**, CLAdapter can be **expected** to be much better than Lora-based methods (**improve 10%+**).
>
>     | **Method** | PACS (Manuscript Table 2, Acc%) | KTH-TIPS2-B (Acc%) | BreakHis (F1%) |
>     |------------|-------------|------------------------------|------------------------------|
>     | LoRA       | 88.53       | 79.18                        | 80.22                        |
>     | DoRA       | 88.43       | 80.28                        | 79.98                        |
>     | MoRA       | 89.09       | 80.11                        | 83.45                        |
>     | **CLAdapter (Ours)** | **91.41** | **91.26** | **93.71** |
>
>
> ---
>
> ### Comment 2:
> > Please provide more analysis on inference efficiency or adapter compression.
>
> ### **Authors’ Response:**
> Thank you for your suggestion. While Figure 4 in the paper already highlights the Fine-tuning efficiency of CLAdapter, we now **additionally** analyze **inference efficiency** and **parameter overhead**.
>
> We evaluate on an NVIDIA 3090 GPU using two typical backbones:
> - ConvNeXt-B + CLAdapter
>   - GFLOPs: 15.42 → 15.86
>   - Params: 88.85M → 99.11M
>   - → Only ~2 FPS drop in inference speed
> - ViT-B + CLAdapter
>   - GFLOPs: 16.86 → 17.86
>   - Params: 85.77M → 92.22M
>   - → Only ~3.5 FPS drop in inference speed
>
> These results demonstrate that CLAdapter **adds minimal inference overhead** while delivering significant gains in diverse data-limited scientific domains.
>
> ---
> ### Comment 3:
> > Please add the dataset/paper reference (e.g. PACS) in the paper body text instead of adding them in the table.
> >
>
> ### Authors’ response:
> Thank you very much for your valued comment, **following your suggestions:**
> - We have revised the manuscript to explicitly mention and cite all benchmark datasets (e.g., PACS, BreakHis, FoliarDisease, KTH, UCF, etc.) **in the main body text** (***Section 4.1**  [Experiment Setup] (**Paragraph** [Downstream Tasks])*, **rather than** referring to them **only in the Table 5/Appendix A.2**.
> - This revision will improve clarity and paper formatting, we appreciate your guidance!

---

### Note · Authors · 2025-08-16

Esteemed Area Chair and Reviewers:

We sincerely appreciate your time and effort in the review process, as well as the valuable feedback, which has helped us refine our work.

- Following the rebuttal phase, we are pleased that all reviewers acknowledged their concerns were satisfactorily addressed and offered encouraging feedback:
  - *“Novel and interesting design”*
  - *“Strong empirical validation”*
  - *“Impactful results”*
  - *“Valuable across data-limited scientific domains”*

- We especially appreciate the following:
  - **Reviewer fsgx** raised their score from 5 to **6** and expressed strong support for acceptance.
  - **Reviewer AwdT** increased their score from 3 to **4** and praised the merit of our work.
  - **Reviewer yjJZ** carefully and first time raised their review and raised the score from 3 to **5**, highlighting the substantial value of our contribution.
  - **Reviewer UMqW**, while not submitting a post-rebuttal update, consistently recognized the novelty, performance, validation, and real-world relevance of our work from the beginning, with a high initial score of **5**.

Finally, we would like to extend our heartfelt thanks once again to all reviewers and the Area Chair for your valuable time, thoughtful feedback, and dedicated efforts throughout the review process.

We look forward to the opportunity to present our work in person at NeurIPS 2025.

---

### Decision · Program_Chairs · 2025-09-17

**Decision:**

Accept (poster)

**Comment:**

This work proposes a method to adapt the representations learnt in large-scale pre-trained vision backbones (e.g., ViT, ConvNext, etc.) to downstream scientific tasks, involving fine-grained discrimination under significant domain shift, and limited training data. A "CLAdaptor" is introduced between the pre-trained backbone and final classification layer which learns to align pre-trained representations with downstream tasks. The adaptor parameters are tuned first, followed by full fine-tuning.

The reviewers appreciated the clarity of writing and strong performance of the proposed adaptor on a number (10) of downstream scientific tasks. Further, clear ablation results wrt the adaptor were presented in the reviewing process.
Some concerns were highlighted: (1) The method is not applied on more 'general' datasets, but limited to the 'scientific' domain; (2) there is no systematic study of performance across varying (or increasing) amounts of training data (on the same dataset / benchmark).

The proposed method, however, achieves strong performance across several benchmarks, and gives a consistent gain over other adaptation strategies (e.g., non-linear adaptation heads, lora, etc.).

In view of the above, the AC is inclined to recommend this work for acceptance. We strongly urge the authors to incorporate the valuable reviewer feedback and the experiments presented in their response in the final version.